# Sensorial and Nutritional Properties of a Collagen-Fortified Snack Bar Designed for the Elderly

**DOI:** 10.3390/nu15163620

**Published:** 2023-08-17

**Authors:** Fatma Hastaoğlu, Emre Hastaoğlu, Nurcan Bağlam, İrem Nur Taş

**Affiliations:** 1Department of Elderly Care, Vocational School of Health, Sivas Cumhuriyet University, Sivas 58140, Turkey; fhastaoglu@cumhuriyet.edu.tr; 2Gerontology Studies Research and Application Center, Sivas Cumhuriyet University, Sivas 58140, Turkey; 3Department of Gastronomy and Culinary Arts, Faculty of Tourism, Sivas Cumhuriyet University, Sivas 58140, Turkey; tasiremnur35@gmail.com; 4Food Studies Research and Application Center, Sivas Cumhuriyet University, Sivas 58140, Turkey; 5Department of Nutrition and Dietetics, Faculty of Health Sciences, Sivas Cumhuriyet University, Sivas 58140, Turkey; nurcanbaglam@cumhuriyet.edu.tr

**Keywords:** elderly nutrition, TOPSIS, product development, collagen, sensory analysis

## Abstract

Background: This study aimed to develop a highly consumable collagen-containing bar that contributes to enriching the diets of elderly individuals, in terms of energy and nutrients. Method: For this purpose, five different bar samples (C, P1, P2, D1, D2) containing different amounts of collagen, date puree, and pumpkin puree were developed and subsequently evaluated in terms of their sensory and nutritional properties by a panel of 30 adult trained sensorial analysists. Results: The bars with the highest flavor score were those with high levels of collagen and pumpkin puree (P2) and date puree (D2). For the analyses of multiple criteria among multiple samples, the TOPSIS technique showed that among the snack bar samples with different contents, the most liked sample was the one with a high level of collagen and date puree (D2). One serving of the developed bars contains approximately 300–400 kcal of energy and 6.8–8.8 g of protein. Considering age-related decreased appetite, as well as chewing and swallowing problems in elderly individuals, regular consumption of nutrient-rich small meals or snacks with enhanced sensory characteristics could contribute to improving nutritional and functional status.

## 1. Introduction

Aging can come with a range of diseases and debilitations, such as cognitive and physical decline, depressive symptoms, and emotional changes, which can directly affect the balance between nutritional needs and intake [1]. Impairment of health, increased use of health services, and mortality are the result of malnutrition and unconscious weight loss [2]. Elderly individuals’ dietary behavior may change due to health or social reasons, decreased taste and smell, or reduced physical strength, which is needed to purchase and prepare food [3]. Furthermore, since the physiology of swallowing changes with advancing age, dysphagia is considered an important health problem in the elderly population. Changes and difficulties in swallowing are also seen in healthy elderly individuals and defined as presbyphagia [4]. In a study by Namasivayam-MacDonald et al., it was reported that 59% of individuals over the age of 65 who stayed in a nursing home for a long time had suspected swallowing disorders [5]. Therefore, to adapt to these changing functions in elderly individuals, choice of foods that are easier to chew comes to the fore.

The main purpose of nutrition planning in the elderly is to provide adequate amounts of energy, protein, micronutrients, and fluids to maintain or improve nutritional status [6]. Experimental and community studies support increased energy and protein intake in the elderly for the maintenance of muscle mass, body function, and overall health [7]. 

Due to decreasing collagen synthesis with aging, collagen supplementation is an inevitable need. The most important source for meeting this need is the consumption of bone and bone broth [8]. However, it is important to specifically observe blood cholesterol levels, as cholesterol levels often rise with aging and foods that are sources of collagen are usually rich in cholesterol as well. Collagen hydrolysates are also produced industrially as an alternative to animal-based foods. They contain high levels of glycine and proline, which are normally deposited in cartilage when properly digested, and are generally considered safe, with minimal known side effects [9]. Collagen hydrolysates in liquid and powder form as food supplements are recommended by experts for both bone and tissue health and skin health. In terms of elderly nutrition, it is possible to add collagen to foods that the elderly consume frequently [10,11]. 

Supplementation of deficient nutrients is a common approach. Furthermore, the use of supplements by malnourished older people could improve vitamin, mineral, or deficient nutrient intakes, but energy, protein, and fiber intakes remain inadequate [12]. However, it is important to consider food or supplements through a highly consumable holistic approach, especially when giving nutritional advice to elderly individuals at risk of malnutrition.

In this study, it was aimed to develop a snack bar in the form of a bar containing collagen with a high level of softness and consumability, and a nutritional value suitable for consumption by the elderly. 

## 2. Materials and Methods

Strained flower honey (production location Zara, Sivas, Turkey), labne cheese (18% milk fat), pasteurized milk cream (18% milk fat), powder corn starch (in pocket), powder sweetened vanillin (in pocket), unflavored powder collagen (in pocket), baking powder (in pocket), milk (3% milk fat), refined sunflower oil, tahini (roasted ground sesame paste), dried dates (mezafeti type), ground oatmeal, roasted hazelnut kernels, roasted pistachio kernels, dried black grapes, walnut kernels, cashews, dried apricots, granulated sugar, wheat flour, eggs (medium), fresh pumpkin, and fresh spinach were purchased from a local market, and natural spring water was used in the study.

### 2.1. Preparation of Samples

The ingredients, recipes, and production stages of the bar samples developed in this study were sensorially determined through preliminary trials executed in our professional application kitchens and the sensorial analysis lab at the Department of Gastronomy and Culinary Arts. In this study, five different bar samples were designed, which all contained oats (19.4 g), roasted hazelnut kernels (1.5 g), roasted pistachio kernels (1.3 g), dried black grapes (1.2 g), walnut kernels (1.2 g), cashew (1.33 g), dried apricots (1.33 g), strained honey (11.25 g), labne cheese (5.55 g), milk cream (5.55 g), granulated sugar (10.1 g); egg (8.88 g), corn starch (0.55 g), flour (7.94 g), vanillin (0.54 g), baking powder (0.55 g), milk (7.27 mL), sunflower oil (4.77 mL), and spinach puree (16.6 g). Table 1 list the ingredients specific to the five bars. Sample C is the “control sample” without collagen, date puree, pumpkin puree, and tahini. Samples P1 and P2 contain tahini, pumpkin puree, and different ratios (x and 2x ratios) of collagen. Samples D1 and D2 contain tahini, date puree, and different ratios (x and 2x ratios) of collagen.

The preparation of bar samples consisted of four stages: preparation of the top layer, preparation of the intermediate layer, preparation of the base layer, and the stages of joining the layers. For all the five snack bar samples, the preparation of components and the construction were the same.

#### 2.1.1. Preparation of the Top Layer of Bar Samples

The nuts (hazelnuts, salted peanuts, black grapes, walnuts, cashews, apricots), oats, and honey were mixed with a whisk in 20 saucepans, poured into a rectangular adjustable cake mold with greaseproof paper on the bottom and baked in a convection oven (Empero, Selçuklu/Konya, Turkey) at 180 °C for 10 min, and the granola layer was prepared.

#### 2.1.2. Preparation of the Intermediate Layer (Puree Layer) of the Bar Samples

Peeled and chopped pumpkin was mixed with water and granulated sugar and kept at 15 °C for 24 h, then cooked in its own juice for about 20 min and pureed with the help of a blender (Sinbo, Istanbul, Turkey), to be used in the interlayers of the P1 and P2 bar samples. When the right consistency of puree was formed, cream was added and mixed to add flavor.

The pitted dates for use in the interlayers of the D1 and D2 samples were cooked in boiling in water and pureed with the help of a blender. When the right consistency of puree was formed, the cream was added and mixed to add flavor.

To make an intermediate layer with the purees, the labneh cheese and granulated sugar were whisked together, then the cream, vanilla, flour, corn starch, and eggs were added. The purees were added to the mixture according to the sample type. 

#### 2.1.3. Preparation of the Base Layer of the Bar Samples

For the preparation of the base layer of the samples, spinach puree was first prepared. Ten pieces of washed and destalked spinach were blanched and pureed with the help of a blender. Using a blender, we whisked the eggs and granulated sugar for 5 min, then added milk, sunflower oil, spinach puree, flour, vanilla, and baking powder, and whisked again. The mixture was poured into a mold with greaseproof paper on the bottom and half-baked in a preheated convection oven at 170 °C for 10 min. Figure 1 shows the final prepared snack bars after baking.

#### 2.1.4. Combining Layers

The intermediate layer mixture was poured onto the base layer after the semi-cooking process and semi-cooked in a steam oven at 160 °C for 15 min. The puree according to the sample type was spread on the cooled bar with the help of a pallet, and the top layer (granola) was placed on it. Thus, our bar samples were prepared for sensory analysis.

### 2.2. Sensory Analysis

The sensory analysis of the bar samples was carried out in two repetitions by a group of 30 panelists (15 male/15 female, average age 22.4) who were trained in nutrition and sensorial analysis and had experience in testing food samples. Panelists were asked to rate the flavor, odor, appearance, and structure of bar samples on a scale of 1 (worst) to 5 (best) [13]. The mean score of each sensory attribute was calculated by averaging the scores obtained from two repetitions [14]. The evaluation was carried out in a sensory analysis laboratory equipped with individual cabinets with temperature control (20–25 °C) and white light. Participants were instructed to drink water between samples, to minimize residual effects.

### 2.3. Ethics Committee Approval

Ethical approval for the study was obtained from Sivas Cumhuriyet University Scientific Research and Publication Ethics Social and Human Sciences Board dated 29 April 2022 and numbered E-60263016-050.06.04-159557.

### 2.4. Application of the TOPSIS Technique for Sensory Analysis 

#### TOPSIS (Technique for Order Preference by Similarity) Method

This method was developed by Hwang and Yoon in 1980 and is a multi-criteria decision making method that has found application in many fields. The evaluation of alternatives (decision options) is based on two basic points: the positive ideal solution, and the negative ideal solution. The TOPSIS method aims to identify the decision option that is the shortest distance from the positive ideal solution and the farthest distance from the negative ideal solution. A positive ideal solution minimizes the cost measure and maximizes the benefit measure. A negative ideal solution is, on the other hand, one that maximizes the cost measure and minimizes the benefit measure. The TOPSIS method reveals distances to positive and negative ideal solutions and also reveals ideal and non-ideal solutions. For the method to be applicable, there must be at least two decision options. With an analysis process that does not involve complex algorithms and mathematical models, the TOPSIS method finds application in many fields, due to its ease of use and easy understanding and interpretation of the results. It has application areas such as personnel selection, supplier evaluation, and selection, location selection, mapping of mineral potentials in mineral deposit research, robot selection, and industry [15].

In this study, the TOPSIS method was employed for evaluation of the sensory properties of snack bars with different ingredients, determining the distance of alternatives from the positive and negative ideal solutions, and also for ranking the treatments. A hierarchy was created by determining the importance of the criteria of flavor, odor, appearance, and structure as 50%, 15%, 20%, and 15%, respectively (Figure 2).

This method is summarized as follows:Construction of the decision-making matrix with *m* (samples) and *n* (flavor, odor, appearance, and structure);Normalization of decision-making matrix;
rij=dij∑k=1ndkj0,,i=1,2,…,n ; j=1,2,…,md.y.Calculate the weights for the criteria and develop the normalized weight matrix *Vij* = *Wij* × *rij*. The weight of each criterion was determined using the Shannon entropy method [16];
$$v=\left[\begin{array}{cccc} w_{1}r_{11} & w_{2}r_{12} & \ldots & w_{m}r_{1m}\\ w_{1}r_{21} & w_{2}r_{22} & \ldots & w_{m}r_{2m}\\ \vdots & \vdots & \vdots & \vdots \\ w_{1}r_{n1} & w_{2}r_{n2} & \ldots & w_{m}r_{nm} \end{array}\right]=\left[\begin{array}{cccc} v_{11} & v_{12} & \ldots & v_{1m}\\ v_{21} & v_{22} & \ldots & v_{2m}\\ \vdots & \vdots & \vdots & \vdots \\ v_{n1} & v_{n2} & \ldots & v_{nm} \end{array}\right]$$Determining the positive and negative solutions;
$$v^{-}=\left\{v_{1}^{-},v_{2}^{-}{,\ldots ,v}_{m}^{-}\right\}\;v^{*}=\left\{v_{1}^{*},v_{2}^{*}{,\ldots ,v}_{m}^{*}\right\}$$Determining the distance of the normalized weighted matrix from the ideal positive and negative points;
$$s_{i}^{*}=\sqrt{\sum_{j=1}^{m}{\left(v_{ij}-v_{j}^{*}\right)}^{2}}\;,\;i=1,2,\ldots ,\;n$$
$$s_{i}^{-}=\sqrt{\sum_{j=1}^{m}{\left(v_{ij}-v_{j}^{-}\right)}^{2}}\;,\;i=1,2,\ldots ,\;n$$Calculating the distance from the ideal point;
$$C_{i}^{*}=\frac{s_{i}^{-}}{s_{i}^{*}+s_{i}^{-}}\;,\;i=1,2,\;\ldots ,\;n$$

### 2.5. Nutritional Analysis

Nutrient analysis of the macro and micronutrients of the bar samples was performed with the Nutrient Database Programme (BeBiS, Ebispro for Windows, Stuttgart, Germany; Turkish Version/BeBiS 8.2). 

### 2.6. Statistical Analysis

The data obtained from the analysis of the bar samples were analyzed using SPSS version 23.0. The statistical significance was set at *p* < 0.05 and a comparison was performed using Tukey’s multiple comparison and ANOVA tests. The bar samples with different formulations were produced twice, and each analysis was repeated three times.

## 3. Results

### 3.1. Sensory Analysis

The sensory values of the snack bars with different ingredients are shown in Figure 3. In terms of appearance, the snack bar sample with low levels of collagen and pumpkin puree (P1) received the highest appearance score, and the snack bar sample with high levels of collagen and pumpkin puree (P2) received the lowest appearance score (Figure 3). It was observed that there was a statistical difference between the appearance values of the samples, but there was no difference between the odor values (*p* > 0.05). 

When the structure values of the snacks to which pumpkin and date purees had been added to provide a softness suitable for consumption by elderly individuals were compared, it was determined that the P1, P2, and D2 samples received the highest appreciation and the difference between the structure values of the samples was statistically significant (*p* < 0.05). 

Flavor is an important criterion in terms of sensory properties and is the most important sensory parameter in industrial food production. Regarding the snack bars containing different ratios of pumpkin and date purees, the highest flavor score was found in the samples containing high levels of collagen, pumpkin puree (P2), and date puree (D2), and there was no statistical difference between the other samples (*p* > 0.05).

When there were multiple criteria among multiple samples, the TOPSIS technique was applied to the samples of snack bars with different contents, and a decision matrix and normalized decision matrix for samples is given in Table 2, while the ranking is given in Table 3. When the study of the taste ranking of the samples was performed, different weighted coefficients were created for the four sensory criteria. Flavor, odor, appearance, and structure criteria were weighted according to the coefficients 0.50, 0.15, 0.20, and 0.15, respectively. According to the ranking results using the TOPSIS technique, which weighted the sensory criteria of the samples (Table 3), the most liked sample was the snack bar (D2) with high levels of collagen and date puree (Figure 3). The ranking of the other samples was determined as the snack bar with a high level of collagen and pumpkin puree (P2), the snack bar with a low level of collagen and date puree (P1), the snack bar with a low level of collagen and pumpkin puree (P1), and the control sample with no collagen and puree (C), respectively.

### 3.2. Nutrients

Table 4 shows the macro and micronutrients in one serving of the snack bars enriched with collagen and shaped into a form that elderly individuals can consume more easily, with different purees. 

The snack bars with the highest energy content compared to the others were snack bars containing date puree (D1 and D2). Samples P2 and D2 had the highest protein content. Considering the amino acids glycine and proline, which play an important role in the structure of collagen, it was shown that sample D2 had the highest content compared to the control sample, which contained no puree.

## 4. Discussion

### 4.1. Sensory Properties

It is known that flavor and sensory properties are the most important factors in food choice [17]. Visual, auditory, and olfactory stimuli influence food intake at all ages. A decrease in the sense of odor and taste changes is caused by the deterioration of sensory functions in the elderly [18,19]. In a study by Braun et al. comparing 30 elderly individuals over 59 years of age with a younger control group in terms of taste and odor, it was shown that the taste and odor scores of elderly individuals were significantly reduced compared to the younger individuals [20]. Seen in up to 50% of elderly individuals, this loss of taste and smell can lead to a loss of appetite [21]. By affecting the hedonistic development of food intake, age-related sensory impairments increase the risk of malnutrition, by causing elderly individuals to give up eating or choose a monotonous diet [22]. 

According to the study by Rusu et al. [23], 55.6% of older people (>60 years old) who took part in the study stated that they were unwilling to eat because of pain and chewing problems, which had a negative effect on their motivation. In the study, meals were designed in the form of a mash or bars that have high values of appearance and taste, in contrast to dysphagia diets that are no longer acceptable in terms of appearance and taste. In relation to the sensory properties of mashed food, 62.5% of the participants indicated that the appearance is the most significant sensory aspect of food. A significant majority of participants, 87.5%, indicated that taste was important or very important. This highlights the importance of offering texture-modified foods that have been specially developed to meet the requirements of older people.

Figure 3 shows the sensory properties of the snack bars with different contents developed within the scope of this study. Flavor is defined as a combination of sensory stimuli produced by consuming food. In addition to affecting taste, sensory properties such as the odor, appearance, and texture of food are considered important determinants of eating behaviors. The bar with the best appearance was the bar sample with low level collagen containing pumpkin puree (P1). The addition of pumpkin (P2) and date puree (D2) to the snack bars developed in this study resulted in a delicious product. In conclusion, it is thought that the snack bars containing pumpkin puree and date puree had a high consumability compared to the others and can provide diversity in the diets of elderly individuals.

In a study conducted by Tsikritzi et al. [24], researchers developed biscuit samples enriched with vitamins and minerals suitable for the consumption of elderly individuals at risk of malnutrition. During the sensory evaluation, it was reported that there were variations in the sensory properties of the micronutrient-fortified biscuits that were produced by incorporating vitamins and minerals. 

Lee et al. [25] reported that there was minimal bitterness due to branched chain amino acid addition (BCAA) in their sensory analysis of chicken curry mousse with BCAA, which they developed for consumption by elderly individuals. In our research, we only added collagen as a supplement, apart from the nutrients inherent in the natural structure of the food. It was decided to add an amount that would not affect the sensory properties during pilot studies for product development.

### 4.2. Nutritional Properties

The risk of malnutrition increases in elderly individuals when oral intake is reduced or in the presence of any risk factor that affects food intake and/or increases nutrient requirements [6]. 

Caregivers need access to ideas and tools that can assist elderly individuals, to help this population overcome the challenges of preparing and consuming a nutritionally rich diet. In addition, one practical way to add nutritious foods to their diets is to prepare simple and nutritious snacks [26]. In a study conducted by Leech et al., among US adults aged 65+, consumption of healthy snacks has been shown to contribute to higher protein, carbohydrate, and fat intake [27]. 

For the elderly, estimated energy expenditure is around 30 kcal/kg, based on variables including gender, nutritional status, physical activity, and clinical status [6]. For the energy content of the bars, it was determined that the D1 and D2 samples had the highest energy content. In a study conducted by Krok-Schoen et al. [28], it was found that the total energy intake from snacks of individuals aged 71 years and elderly individuals during the day was 315 kcal/day, and the total protein intake of the elderly from snacks during the day was 7.2 g/day.

It is thought that these bars, which have an energy content of approximately 300–400 kcal per serving, could be an alternative snack, especially for elderly individuals who have difficulty meeting adequate daily energy requirements. The biggest health concern in the aging population is to slow down the decline in muscle mass and strength and maintain a healthy body weight. Therefore, elderly individuals benefit from a slight increase in dietary protein intake [29,30]. 

Costa and colleagues also stated that their food supplement, specifically designed for the consumption of older individuals, contains 12.7 g of protein per serving (30 g) and provides a significant contribution to meeting their nutritional needs [31]. Regarding to the protein content of the snack bar samples in this study (6.8–8.8 g of protein per serving), protein inclusion in the daily diet of elderly individuals may contribute to supporting protein intake. In a study conducted by Nykänen et al., it was seen that regular use of snacks improved nutritional and functional status in elderly individuals [32]. In addition, in elderly individuals, energy-rich snacks in small volumes are another way to cope with early satiety [33]. Sossen et al. [34] reported in their meta-analysis that fortification practices suitable for elderly nutrition are well-received, particularly among elderly individuals who consume small portions of food and are at risk of nutritional deficiencies. Such fortification practices help increase the intake of energy and nutrients.

Elderly individuals often complain of gastrointestinal problems, including constipation and diarrhea. In addition to its many health benefits, dietary fiber also contributes to improved bowel function. A daily intake of 25 g of dietary fiber is recommended for adults and elderly individuals [35]. According to the Turkish Food Codex Regulation on Nutrition and Health Declarations, food can be defined as “increased/more” when it contains at least 30% more fiber than a similar product [36]. According to this, it can be said that the D1 and D2 samples had increased fiber content compared to sample C. One serving the D1 and D2 bars, which are rich in fiber, could provide up to 17.2% of the recommended daily fiber intake for older adults. 

Studies have reported that protein and micronutrients are among the nutrients associated with health risks related to inadequate intake in elderly [37,38]. However, in the absence of a specific nutrient deficiency, micronutrients can be provided in accordance with the recommendations of the European Food Safety Authority (EFSA) or national nutrition associations for healthy elderly individuals [39]. Kersiene et al. evaluated the micronutrient content of yogurt samples produced for consumption by elderly individuals. The samples were reported to contain adequate levels of vitamin A, vitamin C, vitamin B12, and folic acid [40]. In our study, we evaluated the extent to which the prepared bar samples met the micronutrient requirements of people over 70 years of age using the Turkish Dietary Reference Values [35]. One serving of the bars contributed the most to vitamin A, magnesium, folic acid, and calcium intakes, at 36.9%, 17.6%, 16.7%, and 10.4%, respectively, even though the content of the bar samples varied. It is thought that the snack bars developed within the scope of this study could contribute to micronutrient intake. Aging and poor nutrition can also affect collagen levels in the body. According to studies, functional collagen peptides are effective when taken at levels between 2.5 and 15 g per day [9]. Furthermore, collagen peptides increase mobility, significantly reducing pain in osteoarthritis patients and functional joint pain [41]. However, considering the amount of collagen used in the studies, the snack bars developed in this study had a low collagen content. It is thought that the bar samples may contribute to daily collagen intake when included as a supplement to an adequate and balanced diet, but not alone. 

The fact that the bar samples developed in the study can be produced with easily available products and in the kitchen conditions of institutions such as home/nursing homes, as well as having the potential to be produced and developed commercially, is a strong aspect of the study. Nevertheless, the study has various limitations. One of them is the low amount of collagen in the collagen-enriched bar samples developed. The reason for this was the bitter taste the product had when too much was added during the developmental stages. Another limitation of the study is that the sensory analyses were performed by panelists trained in the field and not by elderly individuals. When the generalizability of the study is evaluated, it can be said that the soft-textured bar samples produced by adding date and pumpkin puree could be consumed by individuals with the chewing difficulties that occur with age. Nevertheless, given the absence of a swallowing safety evaluation, patients diagnosed with swallowing disorders should undergo an individual evaluation by an expert and consume food with a suitable structure based on a swallowing evaluation. Furthermore, it is recognized that nutrient insufficiency, which is often observed in older adults, can be linked with several chronic illnesses. The other limitation of the study is the low generalizability of the developed bar samples to all elderly individuals, due to variations in nutrient requirements and dietary restrictions among elderly individuals with chronic diseases.

## 5. Conclusions

Food fortification is a promising approach to improving the nutritional status of older people. Texture-modified foods fortified with different nutrients can increase food intakes and act as a motivator in the eating behavior of the elderly. Furthermore, fortified products can be developed, but it is crucial to ensure that they taste as similar as possible to their non-fortified counterparts. In this study using pumpkin and date puree, we developed collagen-containing bars with easy consumption and a texture suitable for the elderly. The high energy and protein content, average fiber and collagen content, and ease of consumption due to the date and pumpkin puree make the bars suitable for older people. The study concluded that collagen, along with foods with good nutritional and sensory properties, can be used in recipes or products developed for these populations, to support the well-being of older people. Moreover, these bars could be produced industrially and presented in a way that elderly individuals can easily access.

## Figures and Tables

**Figure 1 nutrients-15-03620-f001:**
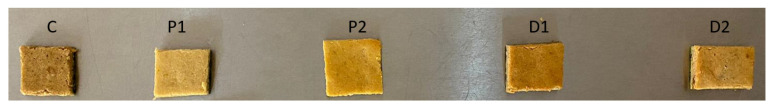
Top view of the base and intermediate layers of the bar samples.

**Figure 2 nutrients-15-03620-f002:**
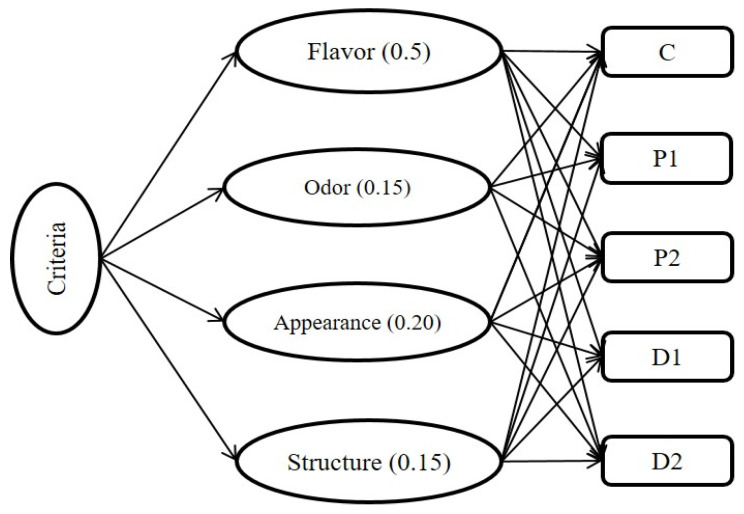
Decision hierarchy process for the snack bars with different ingredients.

**Figure 3 nutrients-15-03620-f003:**
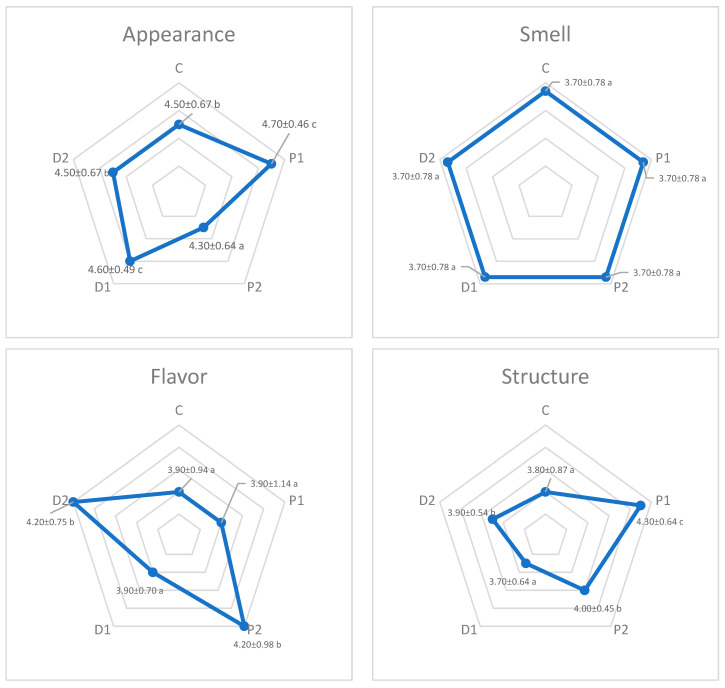
Different sensory properties of the snack bars with different ingredients. Different letters in the same graph mean significantly different (*p* < 0.05).

**Table 1 nutrients-15-03620-t001:** Specific ingredients of the experimental samples.

Material Name	C	P1	P2	D1	D2
Collagen (g)	-	0.33	0.66	0.33	0.66
Pumpkin puree (g)	-	16.6	16.6	-	-
Tahini (ml)	-	6.66	6.66	6.66	6.66
Date puree (g)	-	-	-	10.0	10.0

**Table 2 nutrients-15-03620-t002:** The normalized and weighted decision matrix for snack bars with different ingredients.

**Normalized Decision Matrix**				
**Alternatives**	**Flavor**	**Odor**	**Appearance**	**Structure**
C	04336	0.4542	0.4450	0.4307
P1	0.4336	0.4788	0.4648	0.4874
P2	0.4669	0.4297	0.4253	0.4534
D1	0.4336	0.4419	0.4549	0.4194
D2	0.4669	0.4297	0.4450	0.4421
**Weighted Normalized Decision Matrix**				
**Alternatives**	**Flavor**	**Odor**	**Appearance**	**Structure**
C	0.2168	0.0681	0.0890	0.0646
P1	0.2168	0.0718	0.0930	0.0731
P2	0.2335	0.0644	0.0851	0.0680
D1	0.2168	0.0663	0.0910	0.0629
D2	0.2335	0.0644	0.0890	0.0663

**Table 3 nutrients-15-03620-t003:** The distances (Pi) of each sample obtained by the TOPSIS technique from the positive (D+), negative (D−), and ratio values.

Samples	*Si+*	*Si−*	*Pi*	Ranking
C	0.0195	0.0057	0.2253	5
P1	0.0167	0.0149	0.4713	3
P2	0.0120	0.0174	0.5933	2
D1	0.0204	0.0062	0.2334	4
D2	0.0108	0.0175	0.6185	1

**Table 4 nutrients-15-03620-t004:** Comparison of nutrients in one serving of snack bars with different ingredients.

Nutrients	C	P1	P2	D1	D2
Energy (kcal)	308.3	352.5	353.7	376.4	377.6
Protein (g)	6.8	8.6	8.8	8.5	8.8
Fat (g)	13.1	16.4	16.4	16.4	16.4
Carbohydrate (g)	40.0	42.1	42.1	47.8	47.8
Fibre (g)	3.1	3.8	3.8	4.3	4.3
Vitamin A (µg)	260.4	276.5	276.5	261.0	261.0
Vitamin C (mg)	6.1	13.0	17.9	11.3	16.2
Vitamin E (mg)	4	4.5	4.5	4.3	4.3
Vitamin B6 (mg)	0.1	0.2	0.2	0.2	0.2
Vitamin B12 (µg)	0.1	0.1	0.1	0.1	0.1
Folic acid (µg)	47.1	59.1	59.1	55.2	55.2
Calcium (mg)	64.7	96.3	96.3	99.1	99.1
Phosphorous (mg)	174.6	232.0	232.0	230.7	230.7
Magnesium (mg)	50.0	58.0	58.0	61.7	61.7
Zinc (mg)	1.3	1.9	2.2	1.9	2.2
Glycine (mg)	333.6	511.8	621.7	518.7	628.6
Proline (mg)	431.7	515.1	545.1	524.3	554.3

## Data Availability

The data presented in this study are available on request from the corresponding author. The data are not publicly available because they contain information that could compromise the privacy of the research participants.

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
