# Peer review of "Sensorial and Nutritional Properties of a Collagen-Fortified Snack Bar Designed for the Elderly"

_nutrients, 2023, doi:10.3390/nu15163620_

Round 1

Reviewer 1 Report

This paper shows the results of the development of 5 snack bars that have different food ingredients and are sensorially tested. The developed snack bars differ in puree and collagen content in order to be fortified as well as easier to chew and swallow by the elderly population that is at risk for nutrient deficiencies.

Overall, the results are valuable and well discussed. However, the methods, and textual presentation of the results, should be getting major revisions.

Add to the introduction: aim of the study ‘In order to….’. Actually, the authors should reposition and use lines 232-238 ‘Dysphagia…to the fore’ for this purpose. Or use something similar to lines 273/275: ‘Therefore, this study, it was aimed to develop 273 a snack bar in the form of a bar containing collagen with a high level of softness, consum-274 ability, and nutritional value suitable for the consumption of the elderly.’

Lines 53/57: Can you specify ingredients? Perhaps ecologically corps? Tahini=condiment made from toasted ground hulled sesame, freshly made or purchased already prepared? Purified water=meaning deionized, or something else, can you be more specific?

Omits 59/60 ‘The ingredients, recipes, and production stages of the bar samples developed in the study were sensory determined through preliminary trials executed in our professional dietetics laboratory at the Department of Gastronomy and Culinary’ (or something similar).’  Then continue with this (lines 60/65): ‘In this study, 5 different bar samples were designed that all contained oats (19.4 gr); roasted hazelnut kernels (1.5 gr); roasted pistachio kernels (1.3 gr); dried black grapes (1.2 gr); walnut kernels (1.2 gr); cashew (1.33 gr); dried apricots (1.33 gr); strained Honey (11.25 gr); labneh (5.55 gr); cream (5.55 gr); granulated sugar (10.1 gr); egg (8.88 gr); corn starch (0,55 gr); flour (7.94 gr); vanillin (0.54 gr); baking powder (0.55 gr); milk (7.27 ml); sunflower oil (4.77 ml); and spinach puree (16.6 gr). Table 1 list this ingredients that are specific for the 5 bars. Sample C is the "control sample" without collagen, date puree, pumpkin puree, and tahini. Samples P1 and P2 contain tahini, pumpkin puree and different ratios (x and 2x ratios) of collagen. Samples D1 and D2 contain tahini, date puree and different ratios (x and 2x ratios) of collagen.

Line 87: replace ‘This is a table. Tables should be placed in the main text near to the first time they are cited’ with a title, e.g., ‘Tested snack bar ingredients’

Table 1 only tabulize the ingredients Collagen (gr); Pumpkin puree (gr); Tahini (ml); Date puree (gr). Omit all other ingredients as they can be written in the text in the way under suggestion for Lines 60/65.

Lines 66/68 should read: ‘The preparation of the bar samples consists of 4 stages: preparation of the top layer, preparation of the intermediate layer, preparation of the base layer, and the stages of joining the layers. For all the 5 snack bar samples the preparation of components and the construction are the same.’

Lines 70/71 should read: ‘The nuts (hazelnuts, salted peanuts, black grapes, walnuts, cashews, apricots), oats, and honey were mixed….

Line 89: replace ‘obtained’ with ‘prepared’.

Line 94 should read: ‘in a preheated convection oven at 170°C for 10 minutes. Figure 1 shows the final prepared snack bars after baking.

Line 103 ‘who were trained sensorial analysis’. Perhaps a little bit eelaborate on these individuals: average age, experience in testing food for elderly?

Table 2: can you make bold the highest number per sensation, as to more give the reader a hint on the ranking.

Rephrase Lines 203/204 ‘The most popular sample was the snack bar (D2) with high levels of date puree (Fig. 3).’ Do you mean to say that figure 3 shows the most popular sample? Here only ‘General likes, appearance, smell, flavor and structure is shown. Perhaps, most popular=general likes?

Conclusions should be more specifically promote the benefits (high protein, easy to swallow, average collagen, overall likeable/appreciated snack bar) of the developed collagen/protein bars in this study according to the already mentioned line of thought present in this conclusion section.

Please address the following issues in your re-submmitted paper:

Abstract Lines 17/18 should read: ‘For this purpose, 5 different bar samples (C, P1, P2, D1, D2) con-17 taining different amounts of collagen, date puree, and pumpkin puree, were develop and subsequently evaluated in terms 18 of sensory and nutritional properties by a panel of 30 adult trained sensorial analysists.’

Lines 21/22 should read: ‘For the analyses of multiple criteria among multiple samples the TOPSIS technique showed that among the snack bar samples with different contents, the most liked sample was found to be the one with a high level of date puree (D2).’

Lines 24/26 should read: ‘Considering age-related decreased appetite, chewing and swallowing problems in elderly individuals, regular consumption of nutrient-rich small meals or snacks with enhanced sensory characteristics could contribute to improving their nutritional and functional status.’

Lines 31 should read: ‘debilations’

Lines 32 should read: ‘declines’

Lines 33/34 references needed ‘Elderly….prepare food.’

Lines 35/37 references needed ‘Impairment…..weight loss.’

Lines 45/47 replace ‘However, it is important to specifically observe the blood cholesterol levels, as often with aging cholesterol levels rise, and foods that are sources of collagen usually are rich in cholesterol as well.

Add to the introduction: aim of the study ‘In order to….’. Actually, the authors should reposition and use lines 232-238 ‘Dysphagia…to the fore’ for this purpose. Or use something similar to lines 273/275: ‘Therefore, this study, it was aimed to develop 273 a snack bar in the form of a bar containing collagen with a high level of softness, consum-274 ability, and nutritional value suitable for the consumption of the elderly.’

Methods:

Lines 53/57: Can you specify ingredients? Perhaps ecologically corps? Tahini=condiment made from toasted ground hulled sesame, freshly made or purchased already prepared? Purified water=meaning deionized, or something else, can you be more specific?

Omits 59/60 ‘The ingredients, recipes, and production stages of the bar samples developed in the study were sensory determined through preliminary trials executed in our professional dietetics laboratory at the Department of Gastronomy and Culinary’ (or something similar).’  Then continue with this (lines 60/65): ‘In this study, 5 different bar samples were designed that all contained oats (19.4 gr); roasted hazelnut kernels (1.5 gr); roasted pistachio kernels (1.3 gr); dried black grapes (1.2 gr); walnut kernels (1.2 gr); cashew (1.33 gr); dried apricots (1.33 gr); strained Honey (11.25 gr); labneh (5.55 gr); cream (5.55 gr); granulated sugar (10.1 gr); egg (8.88 gr); corn starch (0,55 gr); flour (7.94 gr); vanillin (0.54 gr); baking powder (0.55 gr); milk (7.27 ml); sunflower oil (4.77 ml); and spinach puree (16.6 gr). Table 1 list this ingredients that are specific for the 5 bars. Sample C is the "control sample" without collagen, date puree, pumpkin puree, and tahini. Samples P1 and P2 contain tahini, pumpkin puree and different ratios (x and 2x ratios) of collagen. Samples D1 and D2 contain tahini, date puree and different ratios (x and 2x ratios) of collagen.

Line 87: replace ‘This is a table. Tables should be placed in the main text near to the first time they are cited’ with a title, e.g., ‘Tested snack bar ingredients’

Table 1 only tabulize the ingredients Collagen (gr); Pumpkin puree (gr); Tahini (ml); Date puree (gr). Omit all other ingredients as they can be written in the text in the way under suggestion for Lines 60/65.

Lines 66/68 should read: ‘The preparation of the bar samples consists of 4 stages: preparation of the top layer, preparation of the intermediate layer, preparation of the base layer, and the stages of joining the layers. For all the 5 snack bar samples the preparation of components and the construction are the same.’

Lines 70/71 should read: ‘The nuts (hazelnuts, salted peanuts, black grapes, walnuts, cashews, apricots), oats, and honey were mixed….

Line 89: replace ‘obtained’ with ‘prepared’.

Line 94 should read: ‘in a preheated convection oven at 170°C for 10 minutes. Figure 1 shows the final prepared snack bars after baking.

Line 103 ‘who were trained sensorial analysis’. Perhaps a little bit eelaborate on these individuals: average age, experience in testing food for elderly?

Results

Line 186 start a new paragraph, that deals with flavor results and start another paragraph dealing with taste (line 190).

Line 199 should read ‘given in Table 2 and

Table 2: can you make bold the highest number per sensation, as to more give the reader a hint on the ranking.

Rephrase Lines 203/204 ‘The most popular sample was the snack bar (D2) with high levels of date puree (Fig. 3).’ Do you mean to say that figure 3 shows the most popular sample? Here only ‘General likes, appearance, smell, flavor and structure is shown. Perhaps, most popular=general likes?

Line 204/Table 3: use Ranking instead of Sorting

Line 217-221 should read: ‘Table 4 shows the macro and micronutrients in 1 serving of snack bars, either enriched with collagen and shaped in a form that elderly individuals can consume more easily through different purees.’ Omit line 221: ‘The nutrients in 1 serving of snack bars with different ingredients are given in Table 221 4.’

Discussion

Lines 273 should read ‘Therefore, this study, was aimed to develop

Lines 276 should read ‘For the energy content of the bars, it was determined that D1 and D2 samples have the highest energy content.’

Conclusions

Should be more specifically promote the benefits (high protein, easy to swallow, average collagen, overall likeable/appreciated snack bar) of the developed collagen/protein bars in this study according to the already mentioned line of thought present in this conclusion section.

Author Response

The recommendations of the Reviewer 1 have been meticulously followed and the article has been revised. 

Reviewer 2 Report

We find the topics tackled in this paper very interesting. However, we feel that the paper as a whole is somewhat lacking in description. The research topic is interesting, but the paper would be better if more diverse studies were cited and compared. We would like to see the content of the paper enhanced and submitted again.

1.Introduction: few references cited. A few more similar studies and the motivation that led to the dispatch of the study should be described and the need for this study discussed.

2. In the discussion, the research of others and the limitations in this study should be described, as well as more on the generalisability of this study.

As there are many aspects throughout the manuscript that are difficult to convey to the reader, the paper would be better if the interpretation and explanation of the results were described in more detail so that researchers with no specialist knowledge could understand them.

Author Response

The recommendations of the Reviewer 2 have been meticulously followed and the article has been revised. 

Round 2

Reviewer 1 Report

The authors adequately addressed all the issues raised in the resubmitted version, that can now be prepared for publication.

Reviewer 2 Report

I confirmed that the points raised have been remedied.

This form is deemed acceptable.